# Nephrotoxicity of Herbal Products in Europe—A Review of an Underestimated Problem

**DOI:** 10.3390/ijms22084132

**Published:** 2021-04-16

**Authors:** Katarzyna Kiliś-Pstrusińska, Anna Wiela-Hojeńska

**Affiliations:** 1Department of Paediatric Nephrology, Wroclaw Medical University, 50-556 Wrocław, Poland; katarzyna.kilis-pstrusinska@umed.wroc.pl; 2Department of Clinical Pharmacology, Wroclaw Medical University, 50-556 Wrocław, Poland

**Keywords:** herbal products, nephrotoxicity, pharmacovigilance

## Abstract

Currently in Europe, despite the many advances in production technology of synthetic drugs, the interest in natural herbal medicines continues to increase. One of the reasons for their popular use is the assumption that natural equals safe. However, herbal medicines contain pharmacologically active ingredients, some of which have been associated with adverse effects. Kidneys are particularly susceptible to injury induced by toxins, including poisonous constituents from medicinal plants. The most recognized herb-induced kidney injury is aristolochic acid nephropathy connected with misuse of certain Traditional Chinese herbal medicines. Data concerning nephrotoxicity of plant species of European origin are scarce. Here, we critically review significant data of the nephrotoxicity of several plants used in European phytotherapy, including *Artemisia herba-alba*, *Glycyrrhiza glabra*, *Euphorbia paralias,* and *Aloe*). Causative mechanisms and factors predisposing to intoxications from the use of herbs are discussed. The basic intention of this review is to improve pharmacovigilance of herbal medicine, especially in patients with chronic kidney diseases.

## 1. Introduction

Currently in Europe, despite the many advances in production technology of synthetic drugs, the interest in natural preparations is increasing. According to the World Health Organization (WHO) data, 80% of the world’s population choose traditional medicines to meet their healthcare needs [1]. Herbal medicines are popular in many countries. They constitute a significant part of the drugs in various therapeutic groups, such as in infectious diseases, rheumatology, or chronic inflammations. One of the reasons for this widespread use is the assumption that they are natural and therefore safe. Furthermore, there are therapeutic areas for which the possibilities of effective treatment have been temporarily exhausted and thus in the public opinion plant substances could be a possible solution. Such a challenge may be directed to a variety of drug-resistant bacterial infections, but also to the unequal fight against viral diseases, and many others, for which today we cannot find an effective cure [2]. However, it should be taken into account that herbal medicines contain pharmacologically active ingredients, some of which have been associated with adverse effects.

The kidneys are particularly susceptible to damage by drugs and toxins [3,4,5]. The mechanisms of kidney injury are different and not yet fully understood. However, the most common pathologic findings in the setting of nephrotoxin exposure were acute tubular necrosis (ATN) and acute interstitial nephritis (AIN) [6]. Over the last two decades publications on nephrotoxicity have increased substantially. They have mainly concerned medications including non-steroidal anti-inflammatory drugs (NSAIDs), antibiotics, diuretics, paracetamol, contrast media, and chemotherapy [7,8].

Among the plants that can cause nephrotoxicity particular attention has been paid to Chinese herbs. “Chinese herb nephropathy” now is appropriately termed aristolochic acid nephropathy (AAN). In the early 1990s in Belgium nine similar cases of rapidly progressive fibrosing interstitial nephritis in young women were described. They were treated with herbs *Stephania tetrandra* and *Magnolia officinalis* in a weight-loss clinic in Brussels. It was later discovered that *S. tetrandra* had been inadvertently replaced by *Aristolochia fangchi* by the manufacturers of the weight-reducing formula. *Aristolochia fangchi* contains aristolochic acid (AA), a plant alkaloid, which is nephrotoxic and carcinogenic in humans and animals [9,10,11,12].

In contrast to AAN, little is known about the nephrotoxicity of plants originally geographically associated with the European distribution area.

Here, we critically review significant data of nephrotoxicity of some European plants recognized as medicinal products both in clinical studies and experimental models. Causative mechanisms and factors predisposing to intoxications from the use of herbs are discussed. The basic intention of this review is to improve pharmacovigilance of herbal medicine, especially in patients with chronic kidney diseases (CKD).

## 2. Mechanism of Plants Nephrotoxicity

There are three main factors involved in increasing exposure to toxins and the risk of kidney injury: high relative proportion of blood flow through the kidneys, their high metabolic activity, and glomerular filtrate reabsorbtion by the renal tubules, which may result in high agent concentration intracellularly. Other mechanisms which may be involved in nephrotoxicity are: hemodynamic alterations, glomerular epithelial cell injury (podocytopathy), and renal inflammation. Some agents can cause hypersensitivity or allergic reaction which manifests as inflammatory infiltrate in the interstitial and tubular compartment (interstitial nephritis) [5,13,14]. It is worth mentioning that adulteration of herbal products with dichromate, cadmium, and phenylbutazone also causes significant renal injury [15].

Knowledge about molecular mechanisms relating to herbal nephrotoxicity is limited although some knowledge can be found from studies on AAN. Aristolochic acid is a generic term that describes a group of structurally related compounds found in the Aristolochiaceae plant family. The consumption of products derived from plants containing AA has been associated with the development of nephropathy and carcinoma.

Several studies using animal models and cultured cell systems have been conducted to understand the molecular mechanisms of AA-induced nephrotoxicity. Some of them are described below. AA may induce oxidative stress by interacting with NADPH oxidase or antioxidative enzymes (GSH, SOD, catalase). Among others, Pozdzick and co-workers demonstrated that AA caused a defective activation of antioxidative enzymes and mitochondrial damage, resulting in tubulotoxicity in the rat model of AAN [16]. In addition, AA induces apoptosis in renal tubular cells through induction of endoplasmic reticulum and mitochondria stress, DNA damage, or activation of the MAPK pathway [17]. AA may also induce apoptosis through inhibition of the PI3K/Akt signaling pathway [18]. Another mechanism related to AA-induced nephrotoxicity is inflammation. As reported by several studies, the acute phase of AAN is followed by persistent interstitial inflammation which leads to fibrosis [19,20]. Fibrosis is a final pathological characteristic of the majority of chronic inflammatory diseases [18]. AA can induce fibrosis via activation of TGF-dependent and c-Jun N-terminal kinase (JNK)/MAP kinase-dependent signaling in mice [19]. Generally, in animal models AA activates tumor necrosis factor (TNF) and transforming growth factor beta (TGF-beta) which trigger a cascade of reactions leading to activation of proinflammatory and profibrotic genes, then leading to inflammation and fibrosis, respectively.

The understanding of the molecular mechanisms of AA-induced nephrotoxicity is of major importance because it might provide a study model for investigating herbal products/drugs-induced kidney toxicity and enable discovery of protective mechanisms against kidney damage.

### Plants Nephrotoxicity and Chronic Kidney Disease

CKD is one of the most important risk factors for deterioration of renal function when using nephrotoxic products. Damaged kidneys have lower capacity to adapt to adverse conditions caused by harmful agents. On the other hand, herbal medical products that disrupt kidney perfusion and are toxic to the renal tubular epithelium belong to independent accelerating disease progression factors. It should be taken into consideration that some groups of patients without evident clinical characteristics of CKD are predisposed to CKD development; for example patients with hypertension, diabetes, heart failure, atherosclerosis, and obesity. Moreover, patients with various kidney diseases use many medications. Interaction of herbal products with conventional drugs is also a potential source of toxicity. In patients with CKD the following is important:-to consider whether the benefits associated with the use of a herbal product significantly outweigh the risk of worsening the excretory function of the kidneys-to take into account interactions between drugs and herbal products (for example: reduction of glomerular filtration, nephrotoxic effects of the drug due to disturbance of its metabolism by plants)-systematically monitor kidney function and control of hydration [21,22,23,24].

In general, nephrotoxicity is connected with many factors which can be categorized into three main groups (Table 1) [3,24]. Most often, at least two factors predispose the development of various forms of clinical kidney disease which may explain the variability and heterogeneity seen in herbal products or toxin-induced kidney disease.

Examples of clinical manifestation of herbal-product nephrotoxicity are shown in Table 2 [25,26,27,28,29,30,31,32,33,34,35,36,37,38,39,40,41,42].

## 3. Nephrotoxicity of European Plant Species

There are no reliable lists of herbs, originally associated with the European geographical area, which may be nephrotoxic. The descriptions of the side effects of plant products available on the European Medicines Agency (EMA) website concern the urinary system in only six cases (see: Table 3) [43,44,45,46,47,48].

Some plants with potential nephrotoxicity are discussed below in more detail.

*Artemisia herba-alba* (family Asteraceae), commonly known as desert or white wormwood, is used in folk medicine as an antihelmintic, antispasmodic, antirheumatic, and antidiabetic agent. The known renal effect of the herb is a diuretic effect [49]. However, *Artemisia herba-alba* can be nephrotoxic. Marrif et al. reported mild hydropic degeneration in proximal convoluted tubules in rabbits and mice orally having an aqueous extract (0.39 g/kg) of *Artemisia herba-alba* [50]. Also Adam et al. described small fatty vacuoles in the cells of the renal proximal convoluted tubules in rats fed on a diet consisting of 10% *Artemisia abyssinica* [51]. The described changes are characteristic for osmotic nephrosis. This term refers to a nonspecific histopathologic finding such as vacuolization and swelling of the renal proximal tubular cells rather than defining a specific entity. Osmotic nephrosis can be induced by many different compounds, such as dextrans and contrast media. It has a broad clinical spectrum that includes acute kidney injury (AKI) and CKD. Aloui et al. published a case of a 59-year-old man with severe acute renal failure with clinical abnormalities typical for tubular injury [52]. Finally, osmotic nephrosis was diagnosed based on kidney biopsy. No causes of AKI were found except of *Artemisia herba-alba* consumption. The patient had taken an aqueous extract of *Artemisia herba-alba*: two cups a day for two consecutive days. The mechanism of kidney toxicity was not clear in the presented case and cause-and-effect relationship was only probable.

*Euphorbia paralias*, the sea spurge, is a species of Euphorbia, native to Europe, northern Africa, and western Asia. The extracts and secondary metabolites from *Euphorbia* plants may act as active principles of medicines for the treatment of many human ailments, mainly inflammation, cancer, and microbial infections. However, the herb may be nephrotoxic. Boubaker et al. described a man with acute renal failure following the ingestion of *Euphorbia paralias*, which was used for treating edema [53]. Acute kidney injury was due to severe tubular necrosis confirmed by kidney biopsy. The exact mechanism of the plant toxicity is not known. However, Euphorbiaceae plants are well known to contain irritant, cytotoxic, and tumor-promoting constituents. The authors suggested that toxic and immunoallergic mechanisms were responsible for the acute renal injury.

*Glycyrrhiza glabra* (licorice). Natural licorice is extracted from *Glycyrrhiza glabra* root containing glycyrrhizin or glycyrrhizic acid. It affects blood pressure and causes other health issues. Licorice may inhibit major renal transport processes needed for filtration, secretion, and absorption [54]. For example, an excessive intake of licorice can cause a hypermineralocorticoidism-like syndrome. Ottenbacher et al. described a case of a 65- year-old woman with previously well controlled hypertension undergoing a licorice-induced hypertensive crisis [42]. She took licorice granules for six months. The patient developed hypertension with sodium retention, edema, hypokalemia, metabolic alkalosis, and low plasma renin activity. Plasma aldosterone and renin activity levels were low. Specific testing was performed, but resolution of symptoms after the patient stopped eating licorice strongly suggested the diagnosis.

Hence, licorice should be carefully monitored for its use in patients, especially in patients with renal problems [21,55,56].

*Cape aloes*. The *Aloe* plant is employed as a dietary supplement in a variety of foods and as an ingredient in cosmetic products. Chemical analysis revealed that the *Aloe* plant contains various polysaccharides and phenolic chemicals, notably anthraquinones. Aloe-emodin is a naturally anthraquinone derivative and an active ingredient of some herbs, such as *Rheum palmatum* L. and *Aloe vera*. Emerging evidence suggests that aloe-emodin exhibits many pharmacological effects, including anticancer, antivirus, anti-inflammatory, antibacterial, antiparasitic, neuroprotective, and hepatoprotective activities. These pharmacological properties lay the foundation for the treatment of various diseases, including influenza virus, inflammation, sepsis, Alzheimer’s disease, glaucoma, malaria, liver fibrosis, psoriasis, type 2 diabetes, growth disorders, and several types of cancers. However, ingestion of *Aloe* preparations may be associated with diarrhea, hypokalemia, and kidney failure [57].

## 4. Risk of Herbal Products Use

In the context of the above information, it is very important to supervise the safety of herbal medicinal products, especially since many people consider them—unlike synthetic drugs—to be free of any side effects, using them contrary to their indications, without consulting a doctor or pharmacist, and often combining them with other medicinal substances, which may lead to undesirable interactions. The risk is increased by the fact that access to scientifically-based knowledge about drugs or herbal substances is not universal, including healthcare professionals, and patients who self-medicate, largely relying on unverified information or scarce available data about their effects and often buying the products via the Internet. The scale of the problem is large because, according to WHO data, 60% of the world’s population and 80% of developing countries prefer herbal medicines [21,58].

The British Medicines Healthcare Products Regulatory Agency (MHRA) drew attention not only to the risks related to the direct toxicity of plant materials, but also, among other problems, to delaying effective pharmacological or surgical treatment, interrupting conventional treatment, as well as the possibility of existing contaminants, which include impurities such as heavy metals, residues of pesticides and fumigants, mycotoxins (aflatoxins, ochratoxin A), microbial contamination, pyrrolizidine alkaloids (PAs), and radioactive substances [29,59,60]. For instance, mercury causes damage to proximal renal tubules and heme biosynthetic pathways, the metal lead to chronic interstitial nephritis, and arsenic to renal tubular necrosis, degeneration and lymphocytic infiltration [30,60]. The natural environment can also be contaminated with cadmium, which means that it also reaches medicinal plants, which are used as herbs (infusions, decoctions), mixtures and herbal preparations, syrups and herbal teas. It was found that the concentration of cadmium in some herbs and herbal preparations, which ranged from 0.033 to 0.306 μg/g of raw material, was higher than the standards permitted by Polish legislation (0.1 μg/g for cadmium). Higher concentrations of cadmium were demonstrated in the preparations of Bronchial, Pektosan, Pulmosan, and Urosan, which may indicate contamination of the raw materials with this element or the special ability of plants included in these mixtures to accumulate cadmium. Since the kidneys are the main accumulation site of cadmium, chronic poisoning with this metal may lead to dysfunction of the proximal renal tubules and the development of tubular proteinuria [61,62]. This problem of heavy metal poisoning is connected especially with child populations consuming herbal medicines. Thirteen cases in USA, UK, Singapore, Hong Kong and United Arab Emirates were identified from 1975 to 2002. In 2003 cadmium toxicity resulting in acute tubulointerstitial nephritis was reported in a 13-year-old girl from Croatia who had consumed a home-made herbal remedy [63].

An additional threat is the growing interest in dietary supplements, which, as products containing medicinal plant raw materials, appeared for the first time in the USA in 1994 on the basis of the Dietary Supplement Health and Education Act (DSHEA). They are, by definition, food products that round off the consumer’s normal diet. At the level of plant raw material quality control, they do not always meet the standards for herbal medicine, which are the principles of Good Agricultural Practice (GAP) limiting pollution caused by the use of plant protection products or heavy metals. For manufacturers, they are a simpler and less costly way to introduce medicinal plant materials to the pharmaceutical market, which does not require clinical confirmation of their effectiveness and safety. In the case of drugs, after their introduction to the market, safety is mandatory, while for supplements there is no such obligation and therefore there is no reliable data. Supplements often contain substances that are ingredients of medicinal products. However, they must then be present in smaller amounts. Most patients, consumers, and even healthcare professionals are unaware of the difference between a herbal medicine and a herbal dietary supplement. The consequence of this ignorance is both the use of dietary supplements in treatments for which they are not intended as well as their abuse [30,64]. For example, cranberry supplements are very popular for prevention of recurrent urinary tract infections, but their excessive ingestion may result in increased urinary excretion of calcium and oxalate, which is conducive to forming calcium oxalate renal stones. They can also negatively affect renal function in patients with pre-existing kidney disease [65].

## 5. Pharmacovigilance of Herbal Medicines

The EMA plays the most important role in the safety-monitoring of medicines in the European Union (EU). It includes the Pharmacovigilance Risk Assessment Committee (PRAC) responsible for assessing and monitoring the safety of human medicines. When safety concerns arise in a medicinal product that has been authorized in more than one Member State, the same regulatory action is taken across the EU, and patients and healthcare professionals in all member states receive the same guidelines. Herbal medicines are subject to the Committee on Herbal Medicinal Products (HMPC). In Poland, the authority appointed to evaluate medicinal products, including herbal products, is the Office for Registration of Medicinal Products, Medical Devices and Biocidal Products (URPLWMiPB). The legal act regulating their presence on the drug market is the Pharmaceutical Law of 6 September 2001 with amendments, while the dietary supplements are regulated by the Act on the safety of food and nutrition of 25 August 2006. The institution supervising the food market, including dietary supplements, is the Chief Sanitary Inspectorate (GIS). The organization that has published many monographs of plant materials used as herbal medicines not only in Europe, but also in other regions, is the WHO. They include, among others, valuable information regarding their safety [66]. Herbal medicinal products can obtain the status of a drug with traditional use registration (Article 16a(1) of Directive 2001/83/EC-European Commission), which means that despite the lack of adequate evidence from clinical trials, their efficacy is reliable and there is evidence of safety in the range for at least 30 years, including at least 15 years within the EU. This status allows for a simplified marketing authorization procedure. Another group are plant products with well-established use of marketing authorization (Article 10a of Directive 2001/83/EC). Scientific literature establishes the fact that the active substances of the medicinal products have been in well-established medicinal use within the EU for at least ten years, with recognized efficacy and an acceptable level of safety. The third possibility is stand-alone or mixed application (Article 8(3) of Directive 2001/83/EC), safety and efficacy data from the company’s own development or a combination of own studies and bibliographic data [67].

In Europe, including Poland, no uniform database containing current toxicological data on plant raw materials has been created so far. In 1992, the document “Listing of herbs and herbal derivatives withdrawn for safety reasons—Herbal drugs with serious risks” was published. In November 2005, the HMPC presented a “public statement on “CPMP (Committee for Proprietary Medicinal Products) List of Herbal Drugs with Serious Risks, dated 1992,” confirming the importance of the information contained therein on the safety of use of the listed plant raw materials, but without its amendment. There is limited information regarding potential adverse reactions associated with herbal administration. Safety data is assessed during the community monograph production process. It is worth emphasizing the approach of Germany, a country with a long tradition of herbal medicine, which publishes official monographs both for plants that are safe, but also for those that pose a threat to humans. These are the so-called monographs of the “E Commission” established by the German authorities to assess the safety and effectiveness of herbal raw materials and herbal medicines [66].

Among those posted on the EMA website, Table 3 includes those that refer to nephrotoxicity. Many, however, contain the information: “The European Medicines Agency is currently developing this information”.

All registered drugs are subject to the obligation of monitoring undesirable effects, i.e., their adverse effects observed with proper use, despite compliance with the contraindications and precautions contained in the medicinal product information, and also the consequences of off-label use, i.e., the use and dosage method inconsistent with the indications approved during the evaluation of the documentation during the registration of the drug, as well as its use contrary to its intended use, abuse, or overdose. The term adverse drug reaction also involves unintentional/inadvertent errors in prescribing, dispensing or administering a medicinal product. Drug-induced complications affect from 3.7% to 38% of patients and, unfortunately, their number is constantly growing. In Europe, they are responsible for around 5% of all hospital admissions. Hospitalized patients also suffer from them. They become the fourth leading cause of death. Adverse drug reactions (ADRs) cause 197,000 deaths per year in the EU. It is estimated that over 2,000,000 serious ADRs occur among all hospitalized patients in the US, which causes more than 100,000 deaths per year. The costs associated with serious side effects in the United States exceed 136 billion dollars annually [68,69]. As stated, up to 44% of drugs approved in the EU are withdrawn from the market due to safety concerns. It is therefore essential that the post marketing safety system, the pharmacovigilance system—which has been defined by the WHO as the science and activities relating to the detection, assessment, understanding and prevention of adverse effects or any other medicine-related problem—works efficiently. The legal framework for pharmacovigilance of medicinal products for human use in the EU is given in Regulation (EC) No 726/2004 and Directive 2001/83/EC on the Community code relating to medicinal products for human use, as amended in 2010 by Regulation (EU) No 1235/2010 and Directives 2010/84/EU and 2012/26 / EU respectively, as well as by the Commission Implementing Regulation (EU) No 520/2012 on the Performance of Pharmacovigilance Activities Provided for in Regulation (EC) No 726/2004 and Directive 2001/83/EC. Nutrivigilance is imposed by the European Food Safety Agency (EFSA) [59,66,70].

Pharmacovigilance and—with regard to herbal medicinal products—phytovigilance, are important elements of health care systems around the world that affect their quality. Unfortunately, compared to other European countries, only a small number of all ADRs reported in Poland is visible. The number of reports from our country entered into the WHO database (VigiBase) as of January 2018 amounted to only 17,345 reports from healthcare professionals, patients, and marketing authorization holders, and in the case of as many as 3873 of them, the report was incomplete. As of April 2020, the VigiBase database contained a total of 46,405 reports submitted from Poland [71].

Every national pharmacovigilance system should pay particular attention to herbs. They need to include information connected with the possibility to identify herbs with a standardized code or nomenclature instead of just brand names. In addition, a not fully efficient Spontaneous Reporting System of obtaining information about the adverse effects of herbs from persons obliged to report them (doctors, pharmacists, nurses, and midwives, paramedics), as well as entitled persons (including patients and their caregivers) makes it difficult to conduct causal analyses. The detection of potential adverse signals of herbal adverse reactions, especially the serious ones that result in death, hospitalization, or significant disability, is challenging, but we should remember that plant-based remedies are not risk-free. Everybody needs to be informed about the possible adverse events, including undesirable effects or interactions with other herbs or synthetic medications [72,73]. Both pharmacokinetic and pharmacodynamic interactions are distinguishable, but identification is difficult because herbal medicines can contain even more than 150 ingredients, and, additionally, they are poorly reported and documented, which is also affected by the lack of information provided by patients to doctors. For example, it has been shown that many flavonoids can alter activity of CYP3A4, CYP2C9, CYP1A2 isoenzymes and in this way interfere with drug metabolism. The information on the website of the Natural Medicines Comprehensive Database may be helpful [74,75].

It should be remembered that too long a use of *Aloe vera* preparations containing aloins leads to electrolyte disturbances, especially hypokalemia. Thiazide drugs, glucocorticosteroids and liquorice, when used in conjunction with aloe, increase the loss of potassium, which is helpful in revealing the toxic effects of glycosides present in foxglove (digitalis). A similar effect is observed with the use of senna. Pseudoaldosteronism caused by the chronic use of liquorice suppresses the renin–angiotensin–aldosterone system. Case reports indicate that long-term use of liquorice causes arterial hypertension with symptoms of encephalopathy, oedema (including pulmonary oedema), hypokalemia, arrhythmias (including cardiac arrest), acute renal failure, weakness of the heart muscle and other muscles, and cardiomyopathy. Liquorice used in conjunction with oral contraceptives by increasing sensitivity to glycyrrhizinic acid may aggravate hypertension, oedema, and hypokalemia. A case of life-threatening hypokalemia-induced muscle paralysis was reported in a patient chronically using liquorice as a sweetener. Nephrotoxicity was observed in patients combining cyclosporine or tacrolimus therapy with long-term use of chamomile tea, which was caused by inhibition of the isoenzyme CYP3A4, which is responsible for the metabolism of immunosuppressants and P-glycoprotein. Increased concentration of cyclosporin and tacrolimus is also observed after the simultaneous intake of drugs or dietary supplements containing milk thistle, ginseng, echinacea, or carambola [65,73,76,77,78]. Hazardous interactions, especially in kidney transplantation patients, may be observed after the combination of cyclosporine or tacrolimus (drugs metabolized by the CYP3A4 isoenzyme) with St. John’s wort, which is one of the inducers of this isoenzyme. It results in a decrease of the immunosuppressive agents concentration and the danger of transplant rejection [66,74,75]. Table 4 shows selected herb–drug interactions, which are manifested by electrolyte imbalance or nephrotoxicity [76,77,78,79,80,81,82].

The current system of monitoring the safety of medicinal products, however, is not sufficient to collect data on all adverse effects associated with the use of natural products, especially since the availability of plant raw materials has significantly expanded recently, and at the same time has reduced the possibility of controlling their use.

Can the risks associated with the use of herbal medicinal products, especially their adverse effects on the kidneys, be reduced? Table 5 presents selected guidelines in this regard [21,30].

In addition remember Theodore Roosevelt’s words: “Risk is like fire: if controlled it will help us; if uncontrolled it will rise up and destroy us” [31].

## Figures and Tables

**Table 1 ijms-22-04132-t001:** Risk factors that increase renal susceptibility to nephrotoxins.

Factors	
Patient-specific factors	kidney disease, other diseases predisposing to kidney injury metabolic disturbances (systemic alkalosis or acidosis, alkaline or acid urine pH), electrolyte abnormalities (hypokalemia, hypomagnesemia, hypercalcemia) older age specific pharmacogenetics (gene mutations in hepatic and renal P450 system, gene mutations in renal transporters and transport proteins) hypovolemia
Kidney-specific factors	high rate of blood delivery high metabolic activity increased toxin concentration in renal medulla and interstitium biotransformation of substances to reactive oxygen species proximal tubular uptake of toxins disruption of proximal tubule polarity leading to glucose accumulation and lipotoxicity
Herbal product-specific factor	direct nephrotoxic effects of the herbal product or its compound herbal-herbal or herbal-drug interaction promoting enhanced nephrotoxicity insolubility of substances and their metabolites in urine prolonged exposure at high doses

**Table 2 ijms-22-04132-t002:** Clinical manifestations of herbal-induced nephrotoxicity (examples).

Symptoms	Plant Species	Literature
Acute kidney injury (AKI)	*Artemisia absinthium* *Glycyrrhiza glabra* *Aloe ferox* *Gloriosa superba* *Colchicum autumnale* *Atropa belladona* *Crataegus orientalis*	[25,26,27,28,29,30]
Direct nephrotoxicity	*Salix daphnoides* *Larrea tridentata Pausinystalia yohimbe* *Salix daphnoides* *Tripterygium wilfordii* *Uncaria tomentosa* *Echinacea angustifolia*	[31,32,33]
Hepatorenal syndrome	*Hedeoma pulegioides* *Thevetia peruviana*	[34,35,36]
Nephrolithiasis	*Vaccinium macrocarpon* *Rheum officinale* *Ephedra sinica*	[32,37,38]
Rhabdomyolysis	*Glycyrrhiza glabra (Licorice)* *Artemisia absinthium*	[30,32,39,40]
Nephrotic syndrome	*Glycyrrhiza glabra, Cyperus rotundus* L., *Citrus* sp. (peel), *Zingiber officinale*	[41]
Hypertension	*Glycyrrhiza glabra*	[42]

**Table 3 ijms-22-04132-t003:** Plant products disturbing kidney function and electrolyte balance listed in EMA monographs.

Latin Name of Source Plant	Latin Pharmaceutical Substance Name	English Name	Side Effects
*Mentha x piperita* L.	*Menthae piperitae aetheroleum*	Peppermint oil	Taken by mouth: an odour of menthol in urine and stools, pain when passing urine, inflammation of the glans of the penis, allergic reactions with headache, slow heart rate, muscle tremor, inability to co-ordinate muscle movements, anaphylactic shock (sudden, severe allergic reaction), contact sensitivity on the mucosa such as the lining of the nose and mouth, and red skin rash, heartburn, burning around the anus, blurred vision, dry mouth, nausea and vomiting
*Rheum palmatum* L.	*Rhei radix*	Rhubarb root	Allergic reactions (such as itching or rash), stomach pain and cramping (sometimes as a result of too high a dose), and diarrhea. Long-term use can lead to reactions including abnormal coloring of the bowel lining, which usually returns to normal when the patient stops taking the medicine, imbalance of salts and water in the body, passage of protein or blood in the urine and yellow or brown coloration of the urine The frequency of these side effects is not known
*Senna alexandrina* Mill.	*Sennae fructus*	Senna pods	Allergic reactions, abdominal (belly) pain, spasm and diarrhea; long-term use may cause coloration of the lining of the intestine, which usually resolves when the patient stops taking the medicine, imbalance of water and salt in the body, presence of proteins and blood in the urine and yellow or red-brown color of the urine
*Rhamnus purshianus D.C*	*Rhamni purshianae cortex*	Cascara	Allergic reactions (such as itching and rash), abdominal (belly) pain, spasm and diarrhea; long-term use can lead to reactions including abnormal coloration of the bowel lining, which usually returns to normal when the patient stops taking the medicine, imbalance of water and salt in the body, presence of proteins and blood in the urine and yellow or red-brown coloration of the urine
*Rhamnus frangula* L.	*Frangulae cortex*	Frangula bark	Stomach pain, spasm and liquid stools, especially in patients with irritable colon, long term use may cause coloration of the gut lining (pseudomelanosis coli) which usually resolves when the patient stops taking the preparation, water and salt imbalance and may result in albuminuria (protein in the urine) and hematuria (blood in the urine), yellow or red-brown (pH dependent) discoloration of urine may occur during treatment
*Aloe vera (L) Burm.f.*	*Aloe ferox* Mill.	Cape Aloes	Stomach and gut disorders such as abdominal (belly) pain and spasm with liquid stools, particularly in patients with irritable colon, long-term use may cause coloration of the lining of the gut which usually returns to normal when the patient stops the medicine, water and salt imbalance and may result in albumin (a protein) and blood in the urine, yellow or red-brown discoloration of urine

**Table 4 ijms-22-04132-t004:** Adverse drug interactions of herbal medicinal products resulting in adverse effects on renal function and electrolyte balance.

Plant Species	Interaction with Plant Substance, Synthetic Drug	Consequences of Interactions
*Aloes folii succus siccatus* Aloe *Aloe barbadensis* Mill. and *Aloe* (various species)	liquorice root, thiazides, and loop diuretics, glucocorticosteroids	exacerbation of electrolyte disturbances, hypokalemia
*Frangulae cortex* Frangula bark *Rhamnus frangula* L.		
*Rhei**radix* Chinese rhubarb *Rheum palmatum* L.		
*Sennae folium* Senna leaf *Cassia senna* L.		
*Liquiritiae radix* Liquorice Root *Glycyrrhiza glabra* L.	diuretics, corticosteroids, stimulant laxatives	increasing water and electrolyte retention, hypokalemia
	thiazides	exacerbation of hypokalemia
*Astragali mongholici radix* Astragalus mongholicus root *Astragalus mongholicus* Bunge	loop diuretics	hypokalemia
*Hippocastani semen* Horse-Chestnut Seed *Aesculus hippocastanum* L.	aminoglycoside antibiotics	risk of nephrotoxicity
*Matricariae flos* Matricaria flower *Matricaria recutita* L. (*Chamomilla recutita* (L.) Rauschert)	cyclosporine, tacrolimus	risk of nephrotoxicity

**Table 5 ijms-22-04132-t005:** How can the risk of nephrotoxicity of plant-derived drugs be reduced?

Use drugs with established knowledge of their pharmacokinetic, pharmacodynamic, and known toxicity.
Pay particular attention to the possibility of their adverse effects on the kidneys, especially while combining them with other drugs and dietary supplements.
Monitor renal function particularly closely while taking drugs with known adverse effects on the kidneys.
Avoid overdosing and prolonged use of herbal drugs.
Report all incidents of adverse effects of these drugs.
Create knowledge bases based on reliable scientific evidence, conduct educational campaigns.
Develop phytovigilance.
Create uniform regulatory standards and develop a strict system of supervision over the safety of medicinal herbal products.
Despite the lack of clinical research for evaluation of herbal medicine renal damage, there is a need for further studies of their nephrotoxicity on a scientific basis.
It is necessary to consider the knowledge of pharmacological aspects of phytotherapy while teaching at medical universities.
Doctors and pharmacists must take the initiative to discuss the adverse reactions of herbal medicines with patients.
Patients should be advised to be alert to possible adverse effects and interactions of herbal medicines.

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
