# Peer review of "Nephrotoxicity of Herbal Products in Europe—A Review of an Underestimated Problem"

_ijms, 2021, doi:10.3390/ijms22084132_

Round 1

Reviewer 1 Report

In this manuscript, Katarzyna Kilis-Pstrusinska and Anna Wiela-Hojenska reviewed the nephrotic toxicity of herb products that’s mainly local to Europe and incorporated those commonly seen in pharmacies or even grocery stores as Rhubarb, Senna and Aloes. In overall, it’s a well-organized review and included the majority of common herb products reported to date, as well as the mechanism of nephrotoxicity. I don’t have major concerns on this review but few minor points as a typo on “intension” on line 57, which I guess would be “intention”.

Author Response

We are grateful to the Reviewer for his/her valuable comments which helped us to improve our manuscript.

Reviewer:

In this manuscript, Katarzyna Kilis-Pstrusinska and Anna Wiela-Hojenska reviewed the nephrotic toxicity of herb products that’s mainly local to Europe and incorporated those commonly seen in pharmacies or even grocery stores as Rhubarb, Senna and Aloes. In overall, it’s a well-organized review and included the majority of common herb products reported to date, as well as the mechanism of nephrotoxicity. I don’t have major concerns on this review but few minor points as a typo on “intension” on line 57, which I guess would be “intention”.

Authors:

We are grateful the Reviewer for his accuracy. We corrected our mistake.

Now it is: “The basic intention of this review is to improve pharmacovigilance of herbal medicine, especially in patients with chronic kidney diseases (CKD)”.

We are grateful to the Reviewer for his/her valuable comments which helped us to improve our manuscript.

Reviewer:

In this manuscript, Katarzyna Kilis-Pstrusinska and Anna Wiela-Hojenska reviewed the nephrotic toxicity of herb products that’s mainly local to Europe and incorporated those commonly seen in pharmacies or even grocery stores as Rhubarb, Senna and Aloes. In overall, it’s a well-organized review and included the majority of common herb products reported to date, as well as the mechanism of nephrotoxicity. I don’t have major concerns on this review but few minor points as a typo on “intension” on line 57, which I guess would be “intention”.

Authors:

We are grateful the Reviewer for his accuracy. We corrected our mistake.

Now it is: “The basic intention of this review is to improve pharmacovigilance of herbal medicine, especially in patients with chronic kidney diseases (CKD)”.

Reviewer 2 Report

This work critically reviews significant data of nephrotoxicity caused by some medicinal plants used in European phytotherapy. In details, mechanism of plants nephrotoxicity and nephrotoxicity of European plant species are discussed, with summaries about the risk factors that increase renal susceptibility to nephrotoxins, clinical manifestations of herbal-induced nephrotoxicity, plant products disturbing kidney function and electrolyte balance listed in EMA monographs, adverse effects on renal function and electrolyte balance resulting from bad drug interactions of herbal medicinal products, and suggestions to reduce the risk of nephrotoxicity of plant-derived drugs. This work may provide some basic knowledge about the nephrotoxicity of herbal medicines in Europe and raise the attention to monitor and control the adverse effect on kidney caused by herbal medicines. Thus, I suggest accepting it for publication, but minor revision is needed before that.

  1. Please clarify the Latin name, English name, Botanical name, Pharmaceutical name, and source of herbal medicines. As far as I am concern, you can keep only the English name, Pharmaceutical name, and source (presented as Latin name) in the manuscript. Also, make sure the typeface (e.g., italics in Latin names) and form of abbreviation are correct, and make it consistent throughout the whole manuscript.
  2. Please define every abbreviation at the first side it occurs.
  3. Every drug, no matter natural or synthetic, may have dual or multiple effects, i.e., pharmaceutical, adverse, and even toxic effects. It is crucial to clarify these effects to provide guidelines for appropriate application of drugs, concerning dose-dependent manner, time-dependent manner, organ-specific action, and disease-specific mechanism, etc. for the deduction of possible causation. Please indicate the administration route, dose, and duration the herbal medicines or herb-derived drugs are taken and compare the toxic dose with the normal dose, where applicable. In addition, please provide other information of herbal medicines/herb-derived drugs about bioavailability, pharmacokinetics, and pharmacodynamics for further reference, if possible.
  4. The length of “Nephrotoxicity of European plant species” is too long. Please rearrange it and provide some sub-tittle to make it clear and easy to read.
  5. Check and revise the reference list regarding format.
  6. Other concerns have been indicated in the PDF file using highlights and notes.

Author Response

Ad Reviewer 2

We are grateful to the Reviewer for his/her valuable comments which helped us to improve our manuscript.

Reviewer:

This work critically reviews significant data of nephrotoxicity caused by some medicinal plants used in European phytotherapy. In details, mechanism of plants nephrotoxicity and nephrotoxicity of European plant species are discussed, with summaries about the risk factors that increase renal susceptibility to nephrotoxins, clinical manifestations of herbal-induced nephrotoxicity, plant products disturbing kidney function and electrolyte balance listed in EMA monographs, adverse effects on renal function and electrolyte balance resulting from bad drug interactions of herbal medicinal products, and suggestions to reduce the risk of nephrotoxicity of plant-derived drugs. This work may provide some basic knowledge about the nephrotoxicity of herbal medicines in Europe and raise the attention to monitor and control the adverse effect on kidney caused by herbal medicines. Thus, I suggest accepting it for publication, but minor revision is needed before that.

  1. Reviewer: Please clarify the Latin name, English name, Botanical name, Pharmaceutical name, and source of herbal medicines. As far as I am concern, you can keep only the English name, Pharmaceutical name, and source (presented as Latin name) in the manuscript. Also, make sure the typeface (e.g., italics in Latin names) and form of abbreviation are correct, and make it consistent throughout the whole manuscript.

Authors: It was made by your suggestion. All herbal names are consistent throughout the whole manuscript.

  1. Reviewer: Please define every abbreviation at the first side it occurs.

Authors: Corrected as suggested.

  1. Reviewer: Every drug, no matter natural or synthetic, may have dual or multiple effects, i.e., pharmaceutical, adverse, and even toxic effects. It is crucial to clarify these effects to provide guidelines for appropriate application of drugs, concerning dose-dependent manner, time-dependent manner, organ-specific action, and disease-specific mechanism, etc. for the deduction of possible causation.

Please indicate the administration route, dose, and duration the herbal medicines or herb-derived drugs are taken and compare the toxic dose with the normal dose, where applicable.

Authors: We have added the missing data (the administration route, dose, and duration the plant administration), if they were given in the cited articles.

In addition, please provide other information of herbal medicines/herb-derived drugs about bioavailability, pharmacokinetics, and pharmacodynamics for further reference, if possible.

Authors:

Traditional herbal medicinal product are used on the basis of a long tradition. According to the information given by producers of herbal drugs: “Pharmacodynamic, pharmacokinetic and bioavailability data are not available as not all active substances are accurately identified”.

  1. Reviewer: The length of “Nephrotoxicity of European plant species” is too long. Please rearrange it and provide some sub-tittle to make it clear and easy to read.

Authors: It was made by your suggestion.

  1. Reviewer: Check and revise the reference list regarding format.

Authors: The reference list has been checked and corrected.

  1. Reviewer: Other concerns have been indicated in the PDF file using highlights and notes.

Authors: According to the Reviewer's remarks, corrections have been made in the REVISION1 version.

Ad Reviewer 2

We are grateful to the Reviewer for his/her valuable comments which helped us to improve our manuscript.

Reviewer:

This work critically reviews significant data of nephrotoxicity caused by some medicinal plants used in European phytotherapy. In details, mechanism of plants nephrotoxicity and nephrotoxicity of European plant species are discussed, with summaries about the risk factors that increase renal susceptibility to nephrotoxins, clinical manifestations of herbal-induced nephrotoxicity, plant products disturbing kidney function and electrolyte balance listed in EMA monographs, adverse effects on renal function and electrolyte balance resulting from bad drug interactions of herbal medicinal products, and suggestions to reduce the risk of nephrotoxicity of plant-derived drugs. This work may provide some basic knowledge about the nephrotoxicity of herbal medicines in Europe and raise the attention to monitor and control the adverse effect on kidney caused by herbal medicines. Thus, I suggest accepting it for publication, but minor revision is needed before that.

  1. Reviewer: Please clarify the Latin name, English name, Botanical name, Pharmaceutical name, and source of herbal medicines. As far as I am concern, you can keep only the English name, Pharmaceutical name, and source (presented as Latin name) in the manuscript. Also, make sure the typeface (e.g., italics in Latin names) and form of abbreviation are correct, and make it consistent throughout the whole manuscript.

Authors: It was made by your suggestion. All herbal names are consistent throughout the whole manuscript.

  1. Reviewer: Please define every abbreviation at the first side it occurs.

Authors: Corrected as suggested.

  1. Reviewer: Every drug, no matter natural or synthetic, may have dual or multiple effects, i.e., pharmaceutical, adverse, and even toxic effects. It is crucial to clarify these effects to provide guidelines for appropriate application of drugs, concerning dose-dependent manner, time-dependent manner, organ-specific action, and disease-specific mechanism, etc. for the deduction of possible causation.

Please indicate the administration route, dose, and duration the herbal medicines or herb-derived drugs are taken and compare the toxic dose with the normal dose, where applicable.

Authors: We have added the missing data (the administration route, dose, and duration the plant administration), if they were given in the cited articles.

In addition, please provide other information of herbal medicines/herb-derived drugs about bioavailability, pharmacokinetics, and pharmacodynamics for further reference, if possible.

Authors:

Traditional herbal medicinal product are used on the basis of a long tradition. According to the information given by producers of herbal drugs: “Pharmacodynamic, pharmacokinetic and bioavailability data are not available as not all active substances are accurately identified”.

  1. Reviewer: The length of “Nephrotoxicity of European plant species” is too long. Please rearrange it and provide some sub-tittle to make it clear and easy to read.

Authors: It was made by your suggestion.

  1. Reviewer: Check and revise the reference list regarding format.

Authors: The reference list has been checked and corrected.

  1. Reviewer: Other concerns have been indicated in the PDF file using highlights and notes.

Authors: According to the Reviewer's remarks, corrections have been made in the REVISION1 version.
